# Complement C3-Deficiency-Induced Constipation in FVB/N-C3^em1Hlee^/Korl Knockout Mice Was Significantly Relieved by Uridine and *Liriope platyphylla* L. Extracts

**DOI:** 10.3390/ijms242115757

**Published:** 2023-10-30

**Authors:** Hee-Jin Song, Ji-Eun Kim, You-Jeong Jin, Yu-Jeong Roh, Ayun Seol, Tae-Ryeol Kim, Ki-Ho Park, Eun-Seo Park, Beum-Soo An, Seung-Yun Yang, Sungbaek Seo, Seong-Min Jo, Young-Suk Jung, Dae-Youn Hwang

**Affiliations:** 1Department of Biomaterials Science (BK21 FOUR Program)/Life and Industry Convergence Research Institute/Laboratory Animals Resources Center, College of Natural Resources and Life Science, Pusan National University, Miryang 50463, Republic of Korea; hejin1544@naver.com (H.-J.S.); prettyjiunx@naver.com (J.-E.K.); hjinyuu1@naver.com (Y.-J.J.); buzyu99@naver.com (Y.-J.R.); a990609@naver.com (A.S.); xofuf0701@naver.com (T.-R.K.); pujihao@naver.com (K.-H.P.); geg9393@naver.com (E.-S.P.); anbs@pusan.ac.kr (B.-S.A.); syang@pusan.ac.kr (S.-Y.Y.); sbseo81@pusan.ac.kr (S.S.); seongmini@pusan.ac.kr (S.-M.J.); 2College of Pharmacy, Pusan National University, Busan 46241, Republic of Korea; youngjung@pusan.ac.kr

**Keywords:** constipation, complement C3, uridine, *Liriope platyphylla* L., mucin, enteric nervous system

## Abstract

Complement component 3 (C3) deficiency has recently been known as a cause of constipation, without studies on the therapeutic efficacy. To evaluate the therapeutic agents against C3-deficiency-induced constipation, improvements in the constipation-related parameters and the associated molecular mechanisms were examined in FVB/N-C3^em1Hlee^/Korl knockout (C3 KO) mice treated with uridine (Urd) and the aqueous extract of *Liriope platyphylla* L. (AEtLP) with laxative activity. The stool parameters and gastrointestinal (GI) transit were increased in Urd- and AEtLP-treated C3 KO mice compared with the vehicle (Veh)-treated C3 KO mice. Urd and AEtLP treatment improved the histological structure, junctional complexes of the intestinal epithelial barrier (IEB), mucin secretion ability, and water retention capacity. Also, an improvement in the composition of neuronal cells, the regulation of excitatory function mediated via the 5-hydroxytryptamine (5-HT) receptors and muscarinic acetylcholine receptors (mAChRs), and the regulation of the inhibitory function mediated via the neuronal nitric oxide synthase (nNOS) and inducible NOS (iNOS) were detected in the enteric nervous system (ENS) of Urd- and AEtLP-treated C3 KO mice. Therefore, the results of the present study suggest that C3-deficiency-induced constipation can improve with treatment with Urd and AEtLP via the regulation of the mucin secretion ability, water retention capacity, and ENS function.

## 1. Introduction

Constipation is a chronic disorder of the gastrointestinal (GI) tract, characterized by infrequent stools and bowel movements, pain and stiffness during defecation, and a sensation of incomplete bowel evacuation [1,2]. There are several well-known causes of constipation including insufficient intake of dietary fibers and water or fluids, lack of physical activity, and the side effects of some medications [3]. There has been an increase in the research on constipation such as therapeutic efficacy and safety studies on new compounds and investigations into the pathogenic mechanism of constipation, because of the rapid increase in the number of patients with constipation [4,5,6]. Animal models with constipation phenotypes have contributed greatly to our understanding of the complex regulatory mechanisms in the human colon [7]. Until now, two induction techniques, through chemicals and diet, were widely applied to produce animal models with constipation phenotypes. In most cases, the human-like constipation was successfully induced by the oral administration or subcutaneous injection of loperamide (Lop) hydrochloride as well as by the feeding of a low-fiber diet containing 41.5% cornstarch, 24.5% milk casein, 10.0% sucrose, 10.0% dextrin, 7.0% mineral mixture, 6.0% corn oil, and 1.0% vitamin mixture for 5 weeks [8,9,10,11]. Also, the phenotypes for constipation could be induced by treatment with opioids and carbon, although these methods are rarely used [12,13]. However, constipation phenotypes caused by alternative regulation of a specific protein or other biological factors have never been used in therapeutic evaluation studies, despite the reporting of new causes of chronic constipation.

Meanwhile, complement component 3 (C3) has been recently investigated as one of the novel causes of chronic constipation based on several indirect indications of a correlation between C3 and constipation, which were detected in cases such as alterative regulation of the C3 protein and mRNA during inflammatory bowel disease (IBD), Crohn’s disease (CD) patients, and ulcerative colitis (UC) patients [14,15,16,17]. Sixteen-week-old C3 knockout (C3 KO) mice showed a significant decrease in parameters associated with stools and evacuation, GI transit, intestinal length, and mucin secretion, as well as remarkable alterations in the histopathological structure and ultrastructure of the mid colon [18]. During the induction of constipation, the inducible nitric oxide synthase (iNOS)-mediated cyclooxygenase-2 (COX-2) induction pathway, inflammasome pathway, and the nuclear factor-kappa B (NF-κB) pathway were activated in the mid colon of C3 KO mice [19]. Also, the excitatory and inhibitory transmissions of the enteric nervous system (ENS) were significantly suppressed, and the distribution of the neuronal cells and intestinal cells of Cajal (ICC) was decreased in C3 KO mice [20]. Furthermore, the C3-deficiency-induced constipation was accompanied by fecal microbiota dysbiosis, which included decreases in the *Anaerocolumna* (97%), *Caecibacterium* (97%), *Christensenella*, *Kineothrix*, and *Oscillibacter* populations, as well as increases in the *Prevotellamassilia* (484%), *Reuthenibacterium* (140%), *Prevotella*, *Eubacterium*, *Culturomica*, *Bacteroides*, and *Muribaculum* populations [21]. However, despite many studies on the molecular mechanism of C3-deficiency-induced constipation, no research studies have been attempted on whether therapeutic products with significant effects in opioids and carbon-induced constipation can effectively improve C3-deficiency-induced constipation.

In this study, two products including uridine (Urd) and the aqueous extract of *Liriope platyphylla* L. (AEtLP) were selected as single chemical drugs and natural products with laxative effects, because the significant effects of them were successfully demonstrated in Lop-induced constipation [22,23]. Especially, the characteristic of Urd as one of the main components of AEtLP will be an advantage to determine their pharmacological effectiveness against another type of constipation disease [24]. Therefore, we investigated whether they can improve the symptoms and regulate the pathogenic mechanism of C3-deficiency-induced constipation. For this, several constipation-related parameters, including parameters associated with stools and evacuation, GI transit, histopathological features of the mid colon, mucin secretion ability, water retention capacity, ENS function, and GI hormone secretion, were analyzed in 16-week-old C3 KO mice after administration of Urd and AEtLP.

## 2. Results

### 2.1. Effects of Urd and AEtLP on Excretion Parameters in C3-Deficiency-Induced Constipation

Firstly, we examined whether Urd and AEtLP, which had proven laxative effects in Lop-induced constipation, can alleviate C3-deficiency-induced constipation. After administration of Urd and AEtLP, the morphology, number, weight and water content of the stools were measured in 16-week-old C3 KO mice with constipation. These parameters showed similar alteration patterns after the administration of Urd and AEtLP, although the rate of changes was varied. The levels of these parameters were lower in C3 KO mice compared with the wild-type (WT) mice prior to treatment with Urd and AEtLP. However, they were significantly increased in Urd- and AEtLP-treated C3 KO mice when compared with the vehicle (Veh)-treated C3 KO mice, which were orally administrated with the same amount of dH_2_O (Figure 1 and Table 1). In particular, the morphology of stools was not fully recovered, and only half of them were observed as normal form (Figure 1). Given that the feeding parameters for all three groups were maintained at constant levels (Table 1), these results suggest that C3-deficiency-induced constipation could be alleviated by the administration of Urd and AEtLP, as observed for Lop-induced constipation.

### 2.2. Effects of Urd and AEtLP on GI Transit in C3-Deficiency-Induced Constipation

The alterations in the intestinal length and the charcoal meal transit were analyzed in C3 KO mice treated with Urd and AEtLP to investigate whether the laxative effects of Urd and AEtLP in C3-deficiency-induced constipation were accompanied by an improvement in the GI motility and length. The total length of the intestine was maintained as a constant in the mice of all the subset groups (Figure 2a,c). However, a significant change was detected in the colon length. The decreased colon lengths in the Veh-treated C3 KO mice were significantly improved in Urd-treated C3 KO mice, although this improvement was not detected in C3 KO mice treated with AEtLP (Figure 2a,d). Meanwhile, the Urd and AEtLP treatments caused significant improvements in the transit ratio of the charcoal meal. This ratio was lower in Veh-treated C3 KO mice than in Veh-treated WT mice. However, it was remarkably increased only in the Urd-treated group (Figure 2a,b). Therefore, the above results indicate that the laxative effects of Urd and AEtLP in C3-deficiency-induced constipation may be closely associated with improvements in GI transit and colon length.

### 2.3. Effects of Urd and AEtLP on the Histopathological Structure of the Mid Colon in C3-Deficiency-Induced Constipation

To investigate whether the laxative effects of Urd and AEtLP in C3-deficiency-induced constipation were accompanied by an improvement in the histopathological structure of the mid colon, the alterations in the mucosa layer, epithelial cells, lamina propria, and muscle layer were analyzed in the hematoxylin and eosin (H & E)-stained mid colons of C3 KO mice after Urd and AEtLP treatments. The thickness of the flat luminal surface, muscle layer, and mucosa layer, as well as the length of the crypt, were lower in the Veh-treated C3 KO mice than in the Veh-treated WT mice. However, these levels were remarkably increased in the C3 KO mice after the administration of Urd or AEtLP compared with the Veh-treated C3 KO mice (Figure 3 and Table 2). Also, significant alterations were detected in the epithelial cells, including enterocytes, paneth cells, and goblet cells. Alterations in the shape, number, and irregular arrangement of the epithelial cells were restored by the Urd and AEtLP treatments (Figure 3 and Table 1). Furthermore, the increase in the muscularis mucosae thickness in the Veh-treated C3 KO mice was decreased only in the Urd-treated C3 KO mice, although it was maintained at a constant in the AEtLP-treated C3 KO mice (Figure 3 and Table 1).

We also analyzed the alterations in the expression level of junctional proteins in the junctional complexes to examine whether the improvement seen with Urd and AEtLP treatment on the histopathological structure of the mid colon of the C3 KO mice was accompanied by the regulation of the two junctional complexes between the epithelial cells. Among the components of the tight junction, the transcription levels of zonula occludens-1 (ZO-1), occludin, claudin-1, and claudin-4 were lower in the Veh-treated C3 KO mice than in the Veh-treated WT mice. However, the transcription levels of ZO-1 and claudin-1 were significantly increased in the Urd- or AEtLP-treated C3 KO mice, while those of occludin and claudin-4 were maintained at a constant level in both groups (Figure 4a). Furthermore, the transcription level of p120 for the adherens junction was increased in the C3 KO mice after administration of only AEtLP (Figure 4b). Therefore, these results suggest that the laxative effects of Urd and AEtLP in C3-deficiency-induced constipation may be associated with an improvement in the histopathological structure and junctional complexes in the mid colon of C3 KO mice.

### 2.4. Effects of Urd and AEtLP on Mucin Secretion and Water Retention in C3-Deficiency-Induced Constipation

Next, we measured the levels of mucin secretion and the expression of the membrane water channel in the mid colon of the Urd- and AEtLP-treated C3 KO mice to determine whether the laxative effects of Urd and AEtLP in C3-deficiency-induced constipation were accompanied by an alteration in the mucin secretion ability and water retention capacity. Mucin secretion, as observed on the Alcian-blue-stained section, was lower in the Veh-treated C3 KO mice than in the Veh-treated WT mice. However, these levels were remarkably increased in the mid colon of the C3 KO mice after administration of Urd and AEtLP (Figure 5a). Also, these alterations were completely reflected in the expression levels of the genes associated with mucin secretion. The Urd and AEtLP treatments induced the upregulation of mucin-1 (MUC1), MUC2, and Kruppel-like factor 4 (Klf4) transcription in the mid colon of C3 KO mice, although the rate of increase in their level was varied (Figure 5b). Furthermore, the transcription levels of aquaporin 3 (AQP3) and AQP8 genes were also observed to have a similar pattern in both Urd- and AEtLP-treated C3 KO mice. The decrease in the expression of these genes in the C3 KO mice increased after the Urd and AEtLP treatments, although this increase varied between the two treatments (Figure 6). Taken together, these results indicate that the laxative effects of Urd and AEtLP in C3-deficiency-induced constipation may be associated with an improvement in the mucin secretion ability and water retention capacity.

### 2.5. Effects of Urd and AEtLP on ENS in C3-Deficiency-Induced Constipation

To investigate whether the laxative effects of Urd and AEtLP in C3-deficiency-induced constipation were accompanied by an improvement in the ENS function, the alteration in the composition of neuronal cells and the excitatory and inhibitory function of the ENS were measured in the mid colon of C3 KO mice treated with Urd and AEtLP. First, the density of the neuronal cells and ICC were determined based on the levels of expression of the key markers for these cells. The expression levels of receptor protein kinase kit (C-kit), neuron-specific enolase (NSE), and protein gene product 9.5 (PGP9.5) proteins were lower in the Veh-treated C3 KO mice than in the Veh-treated WT mice. However, these levels significantly increased in the Urd- and AEtLP-treated C3 KO mice compared with the Veh-treated C3 KO mice (Figure 7). Also, an alleviation of the excitatory function of the ENS was determined by the level of the regulatory factors for 5-hydroxytryptamine (5-HT) and muscarinic acetylcholine receptors (mAChRs) function. The transcription levels of four 5-HT receptors, including the 2AR, 2BR, 3AR, and 3BR, were remarkably enhanced in the Urd- and AEtLP-treated C3 KO mice, although the highest rate was detected in the 5-HT 3BR level (Figure 8). Furthermore, a similar regulation pattern was observed in the signaling pathway of mAChR downstream. The increased level of acetylcholinesterase (AChE) activity in the Veh-treated C3 KO mice was remarkably decreased after the administration of Urd and AEtLP (Figure 9a). Among the two mAChRs, a significant increase was detected in the expression of mAChR M2 after the Urd and AEtLP treatment, while that of mAChR M3 remained constant in the same group (Figure 9b). The expression level of the G-protein alpha subunit (Gα), a key member of the mAChR downstream signaling pathway, was higher in the Veh-treated C3 KO mice than in the Veh-treated WT mice. This level was remarkably decreased in the Urd- and AEtLP-treated C3 KO mice compared with the Veh-treated C3 KO mice (Figure 9c). Moreover, alterations in the expression of two enzymes in the downstream pathway, protein kinase C (PKC) and phosphoinositide 3-kinase (PI3K), were reflected in the regulation of the Gα expression. Their levels in the C3 KO mice were decreased after the administration of Urd and AEtLP (Figure 9d). A similar regulatory pattern was detected in the phosphorylation of extracellular signal-regulated kinase (ERK), p38, and NF-κB inhibitor alpha (IκB-α). The levels of these three proteins were significantly decreased in the Urd- and AEtLP-treated C3 KO mice compared with the Veh-treated C3 KO mice (Figure 9e).

In addition, the effects of the Urd and AEtLP treatments were observed in the inhibitory function of the ENS. The enhanced nitric oxide (NO) concentration and the expression level of iNOS in the Veh-treated C3 KO mice were remarkably decreased after the administration of Urd and AEtLP. However, the reverse was observed for the NO concentration and neuronal NOS (nNOS) expression in the neurons. After administration of Urd and AEtLP, the expression of nNOS increased in the C3 KO mice (Figure 10a,b). Taken together, the above results indicate that the laxative effects of Urd and AEtLP in C3-deficiency-induced constipation may be closely associated with an improvement in the ENS function.

### 2.6. Effects of Urd and AEtLP on the Regulation of the GI Hormone in C3-Deficiency-Induced Constipation

Finally, we analyzed the alteration in the concentration of GI hormones in the mid colon of Urd- and AEtLP-treated C3 KO mice to determine whether the effects of Urd and AEtLP in improving the ENS function during C3-deficiency-induced constipation were accompanied by an alteration in the regulation of the GI hormones. The concentration of two key hormones, cholecystokinin (CCK) and gastrin, were lower in the Veh-treated C3 KO mice than in the Veh-treated WT mice. However, their concentrations were significantly enhanced in the Urd- and AEtLP-treated C3 KO groups compared with the Veh-treated group. Specifically, the highest increase in the CCK concentration was detected in the Urd-treated C3 KO mice (Figure 11). Therefore, the improvement in the ENS function through the Urd and AEtLP treatments during C3-deficiency-induced constipation is reflected in the regulation of CCK and gastrin concentrations.

## 3. Discussion

Several natural products and substances, including *Ecklonia cava*, *Liriope platyphylla* L., *Asparagus cochinchinensis*, *Galla Rhois*, *Spicatoside A*, Urd, and quercetin have been investigated as medicinal compounds with laxative effects, but only against Lop-induced constipation [23,25,26,27,28,29,30]. However, they have not been applied to the therapeutic efficacy against constipation caused by various biological factors in order to expand their potential for all types of constipation. In the present study, the laxative effects of Urd and AEtLP were examined in C3 KO mice with C3-deficiency-induced constipation to investigate whether substances that have therapeutic effects on Lop-induced constipation are also effective in C3-deficiency-induced constipation. These results provide novel scientific evidence that the administration of Urd and AEtLP improves constipation-related symptoms and parameters in C3-deficiency-induced constipation through the regulation of mucin secretion, ENS function, and GI hormones. They suggest that these two substances can also work effectively on Lop-induced constipation as well as C3-deficiency-induced constipation at the same time. However, further studies to determine the optimal dosage and mechanism of action are warranted.

Generally, Urd is known to play a key therapeutic role in many physical and biochemical processes such as neuroprotection, biochemical modulation, the glycolysis pathway, and mitigation of side effects of medical drugs [31,32,33,34]. Also, the therapeutic effect of this compound was additionally detected in Lop-induced constipation in an earlier study [23]. These effects were analyzed in general constipation phenotypes including stools excretion, mucin secretion, and histopathological changes of the colon. In the Lop-injected mice, key stool parameters including the number, weight, and water content, which were remarkably increased by 85%, 105%, and 110%, respectively, after the administration of Urd [23]. In the present study, in the C3-deficiency-induced constipation model, Urd administration induced similar laxative effects on the stool parameters. These parameters were increased by 129%, 82%, and 159%, respectively, in Urd-treated C3 KO mice. These results show that the administration of Urd can induce similar effects of improving the excretion of stools in both Lop-induced and C3-deficiency-induced constipation. In addition, similar effects of Urd were detected in the mucin secretion and expression of AQP, as well as in the expression of PI3K and PKC. The C3-deficiency-induced constipation model showed a 140% and 118% increase in these factors, respectively, while they were increased by 150% and 513%, respectively, in the Lop-induced constipation model [23]. The levels of PI3K and PKC were similarly downregulated in the Lop-induced constipation model (−43.6% and −92.8%) and C3-deficiency-induced constipation model (−11.7% and 23%) after Urd administration [23]. Especially, the remarkable reduction (−92.8%) in the PKC level after Lop administration is likely to be closely related to the function of Lop as a calmodulin system blocker in the colon due to PKC cross-talk with multifunctional Ca^2+^/calmodulin-dependent protein kinase [35,36]. However, significant differences in the histopathological structure of the colon due to the Urd treatment were observed between both groups. The thickness of the mucosa and muscle layer was remarkably enhanced by 275% and 221%, respectively, after the administration of Urd in the Lop-induced constipation model [23], while these levels were slightly increased (7%) or decreased (-39%) in the C3-deficiency-induced constipation model. Overall, these results suggest that the effects of Urd in enhancing the histopathological structure in the mid colon of rodents were better in the case of Lop-induced constipation than in C3-deficiency-induced constipation (Table 3). These differences may be due to the differences in the animal species used in the two experiments, because the Lop-induced constipation study used Sprague–Dawley (SD) rats, and the C3-deficiency-induced constipation study used FVB mice. Until now, several strains of rodents, including SD rats, Wistar rats, Balb/c mice, Institute of Cancer Research (ICR) mice, and ddY mice, have been widely used as Lop-induced constipation models to examine the laxative effects of some natural products and chemicals [22,37,38,39,40]. However, the optimal conditions for the induction of constipation varied from species to species [41].

Meanwhile, the therapeutic effects of Urd in Lop- and C3-deficiency-induced constipation is thought to be closely related to G proteins that act as molecular switches for intracellular signaling cascades and transmitters of extracellular signals derived from outside environments [42]. Firstly, two constipation-inducing mechanisms are linked to the action of G-protein-coupled receptors (GPCRs) on the plasma membrane. Lop bind to the μ-opioid receptors as opioid-receptor agonists in the myenteric plexus of the large intestine. The signal induced by the binding of Lop and opioid receptors is mediated by stimulating heterotrimeric G proteins [43]. Also, a similar alteration mechanism on activation of G proteins was detected during C3-deficiency-induced constipation. The normal function of the gastrointestinal tract may be suppressed through the dysregulation of C3a receptor mediated signaling pathways due to this receptor being one of the members of the GPCRs [44,45,46]. Furthermore, alterations on the receptor signaling pathway for two neurotransmitters, 5-HT and muscarinic acetylcholine, was investigated during the investigation of the therapeutic effects of Urd in the constipation model. G proteins play a key role in the downstream signaling pathway of these two receptors [47,48]. The urd treatment induced some significant alterations on the 5-HT receptor signaling pathway during the improvement of Lop-induced constipation. The levels of Gα expression and PKC phosphorylation were recovered by Urd in Lop-pretreated epithelial and neuronal cells [24]. Also, the downstream signaling pathway of mAChR M2 and M3, including the expression of mAChR M2 and M3, as well as the phosphorylation of PKC and PI3K proteins, were significantly recovered in the Lop-induced constipation model after Urd treatment [23]. In the present study, the laxative mechanism of Urd was similarly detected in C3-deficiency-induced constipation, although there are a few differences in the system of analyses [23]. The 5-HT receptor and mAChR signaling pathway were remarkably recovered in the case of C3-deficiency-induced constipation after Urd treatment. But the expression levels of mAChR M3 showed a reverse pattern between Lop-induced constipation and C3-deficiency-induced constipation.

*Liriope platyphylla* L. is a well-known traditional herb that has exhibited excellent therapeutic effects in the management of various chronic diseases including asthma, atopic dermatitis, neurodegenerative disease, obesity, and diabetes [49,50,51,52,53]. These effects are thought to be due to the composition of *Liriope platyphylla* L., although their ingredients were completely identified. The roots of this plants contain various compounds and substance such as polyphenolic compounds, diosgenin (DG), 5-hydroxymethylfurfural (5-HMF), adenosine (AD), hydroxypropyl cellulose (HPC), Urd, (-)-Liriopein B, LP-A, LP-B, lupenone, and beta-sitosterol [54,55,56,57,58]. However, the chemical composition of AEtLP should be further analyzed, because bioactive components derived from plants can be affected by multiple factors including growth conditions, time of harvest, post-harvest treatment, and natural variability. Also, the novel therapeutic effects of the *Liriope platyphylla* L. extract on chronic constipation have been investigated in the Lop-induced constipation model [22,24,59]. AEtLP administration to Lop-induced constipation SD rats remarkably increased the number, weight, and water contents of stools by 67%, 96%, and 122%, respectively [22]. Similar effects on the stool parameters were detected in the present study on C3-deficiency-induced constipation. However, the effects of AEtLP on the total weight and water content of stools were greater in the case of C3-deficiency-induced constipation than in Lop-induced constipation. Especially, AEtLP administration induced a significant increase in the stool number and water contents in WT mice, as shown in Table 1. These alterations are thought to be linked to the functions of the five components (DG, 5-HMF, AD, HPC, and Urd) that make up the AEtLP [22]. Also, the effects of AEtLP on the histopathological structure of the colon were observed to be common in both the Lop-induced constipation and C3-deficiency-induced constipation models, except for the effect on the muscular layer thickness [22]. However, it was not possible to directly compare the structural characteristics of epithelial cells and crypt, as well as GI motility and length, because of the differences in the analysis factors considered in each study. Moreover, the effects of Urd on the downstream signaling pathway of mAChR were observed to be similar in both the C3-deficiency-induced constipation and Lop-induced constipation models, although there were a few variations in some factors [22,24] (Table 3). Therefore, it can be seen that AEtLP has similar therapeutic effects in improving chronic constipation induced either by Lop injection or C3 deficiency. Thus, these results suggest that AEtLP has potential as a treatment for constipation of various etiologies.

Moreover, this study has a few limitations compared with the cited previous studies, which analyzed the therapeutic effects of natural products and compounds in Lop-induced models, since the studies reported by other research teams are not cited. However, the citations in this study were sufficiently reasonable to compare the constipation phenotypes between studies, because most previous results using Lop-induced models have been published by our team [23,26,27,29]. Also, the important information on the role and action mechanism of C3 deficiency on the constipation of C3 KO mice was only provided by our team [18,19,20,21]. Therefore, this study is not simply conducted to compare the efficacy of drug candidates for constipation in both animal models, but to evaluate the possibility of new treatments for constipation caused by other biological causes. Furthermore, another limitation was that our study investigated only two candidates as treatments for C3-deficiency-induced constipation. Additional studies are needed to identify various natural products and chemical compounds applicable to this disease.

## 4. Materials and Methods

### 4.1. Preparation of Urd and AEtLP

AEtLP for administration to C3 KO mice was prepared as described in our previous studies [22]. After deposition of the dry root samples of *Liriope platyphylla* L. as voucher specimens (WPC-11-010) at the Functional Materials Bank of the PNU-Wellbeing RIS Center at Pusan National University (PNU, Miryang, Republic of Korea), AEtLP was extracted from a mixture of the dry root powder of *Liriope platyphylla* L. and distilled water (dH_2_O) (600 g: 2 L ratio) at 60 °C for 2 h using circulating extraction equipment (IKA Labortechnik, Staufen, Germany). Also, Urd was provided by Sigma-Aldrich Co. (Saint Louis, MO, USA), as described in previous study [23].

### 4.2. Care and Use of Animals

The protocols for the study of the efficacy of Urd and AEtLP in animal models were performed with the approval of the PNU-Institutional Animal Care and Use Committee (PNU-IACUC, Busan, Republic of Korea) for review of animal experimentation ethics and science (Approval number: PNU-2021-0098). All the mice used in the experiment were managed at the PNU-Laboratory Animal Resources Center, which was accredited by the Korea Food and Drug Administration (KFDA, Cheongju, Republic of Korea) (Accredited Unit Number; 000231) and the Association for Assessment and Accreditation of Laboratory Animal Care (AAALAC, Frederick, MD, USA) International (Accredited Unit Number; 001525). The mice were managed in a specific pathogen-free (SPF) state with a 12 h light cycle (on at 8:00 h; off at 20:00 h) at a temperature of 23 ± 2 °C and a relative humidity of 50 ± 10%. They were provided with filtered tap water and an irradiated standard chow diet (Samtako BioKorea Co., Osan, Republic of Korea) *ad libitum*. Also, the animal care and use for this protocol was conducted as described in previous studies based on NIH Guide for the Care and Use of Laboratory animals [60].

### 4.3. Experimental Design for C3 KO Mice

WT and C3 KO mice with the FVB/N Korl background were produced by mating heterogenous type (HT) (male) and HT (female) mice and subsequently typed using specific primers, as described in previous studies (Appendix A) [18]. After selecting mice with constipation phenotypes from among the 16-week-old C3 KO mice based on a decrease in the stool parameters, WT (*n* = 24) and C3 KO (*n* = 24) mice of the same age were allocated to one of three groups; Veh-treated group (Veh-treated WT mice (*n* = 8) and Veh-treated C3 KO mice (*n* = 8)), Urd-treated group (Urd-treated WT mice (*n* = 8) and Urd-treated C3 KO mice (*n* = 8)), and AEtLP-treated group (AEtLP-treated WT mice (*n* = 8) and AEtLP-treated C3 KO mice (*n* = 8)). The Veh-treated group was orally administrated with the same volume of dH_2_O, while the Urd-treated group and the AEtLP-treated group were orally administrated 100 mg/kg of Urd and 1000 mg/kg of AEtLP at one time. The dosages for Urd and AEtLP treatment of the C3 KO mice were decided based on the results from previous research on the laxative effects of Urd [23] and AEtLP [22]. Also, human-equivalent doses (HED) were calculated based on the body surface area using the formula for dose translation [61]. Therefore, a 100 or 1000 mg single dose of Urd or AEtLP in mice is equivalent to an 8.1 mg or 81 mg daily dose in humans, respectively. At 24 h after the final administration, the WT and C3 KO mice were euthanized by a trained researcher using an appropriate chamber with a gas regulator and CO_2_ gas with a minimum purity of 99.0% based on the American Veterinary Medical Association (AVMA, Schaumburg, IL, USA) Guidelines for the Euthanasia of Animals. A cage containing mice was placed in the chamber and CO_2_ gas of 99.0% was introduced into the chamber without precharging, with a fill rate of ~50% of the chamber volume per minute. The final death of mice was confirmed by ascertaining cardiac and respiratory arrest or dilated pupils and fixed bodies. The mid colon samples were collected from mice of the subset group for histopathological analyses and molecular assay.

### 4.4. Measurement of Food Intake and Water Consumption

Food and water collected from the WT and C3 KO mice in the metabolic cage (Daejong Instrument Industry Co. Ltd., Seoul, Republic of Korea) were weighed using a chemical balance and measuring cylinder. The levels of food intake and water consumption per day were calculated based on these data. All measurements were performed twice for accuracy.

### 4.5. Measurement of Excretion Parameter

The levels of excretion parameters including stools and urine were measured as described in previous studies [9,18]. Briefly, all mice were bred in metabolic cages (Daejong Instrument Industry Co. LTD) to obtain uncontaminated stools and urine samples. The total stools of each mouse were harvested from the metabolic cage at 9 a.m., and the weight of the stools was measured twice using a chemical balance. The water content of the stools was analyzed as follows:Stool water content (%) = (A − B)/A × 100

Here, A is the weight of fresh stools, and B is the weight of stools after drying at 60 °C for 12 h. Also, the urine volume was measured immediately after being collected, twice per sample, using a cylinder.

### 4.6. Analysis of GI Motility and Length

GI motility and length were measured using the methods described in previous studies [8,9]. At 24 h after the Urd and AEtLP administration, WT and C3 KO mice were fed a 0.5 mL charcoal meal (3% suspension of activated charcoal in 0.5% aqueous methylcellulose) (Sigma-Aldrich Co.). After 30 min, all the mice were sacrificed with CO_2_ gas, and the complete GI tract was collected from the abdominal cavity. The total intestine length from the stomach to the anus and the transit distance of the charcoal meal were measured using a ruler. The charcoal transit ratio was calculated as follows:Charcoal meal transit ratio (%) = [(Total intestine length-Transit distance of charcoal meal)/Total intestine length] × 100.

Also, the length of the colon and intestine was measured twice using a ruler.

### 4.7. Histopathological Analysis

The mid colon samples were collected from the WT and C3 KO mice for H & E and Alcian blue staining analysis. Briefly, the collected tissues were fixed in 10% formalin for 48 h, embedded in paraffin wax, sectioned into 4 µm thick slices, and prepared on a glass slide. For H & E staining analysis, the mid colon sections were dipped into 4% H & E solution (Sigma-Aldrich Co.). The morphological characteristics, including the thickness of the muscle and mucosal layer, as well as changes in the mucus layer, epithelial cells, and lamina propria for intestinal epithelial barrier (IEB) abnormality, were observed in the mid colon under an optical microscope and analyzed using the Leica Application Suite (Leica Microsystems, Heerbrugg, Switzerland).

For mucin staining analysis, the mid colon sections were stained using an Alcian blue staining kit (IHC world, Woodstock, MD, USA), and their morphological characteristics were observed under an optical microscope (Leica Application Suite, Leica Microsystems). The mucin intensity was measured using the Image J program 1.52a (NIH, Bethesda, ML, USA) together with Image-Color-Split Channels and the Analysis-Tools-ROI Manager.

### 4.8. Western Blot Analysis

Western blot analyses to examine alterations in the protein expression were performed as described in previous studies [62,63]. For extracting the tissue homogenate, the Pro-prep Protein Extraction Solution (iNtRON Biotechnology Inc., Seongnam, Republic of Korea) was used under homogenization with a Polytron PT-MR 3100 D Homogenizer (Kinematica AG, Lusern, Switzerland). After extracting the supernatant using centrifugation at 13,000× *g* for 5 min, the protein concentration was confirmed using the Bicinchoninic acid Protein Assay (BCA) Kit (Thermo Fisher Scientific Inc., Waltham, MA, USA). The protein (30 µg) was separated via 4–20% dodecyl sulfate-polyacrylamide gel electrophoresis (SDS-PAGE) for 2 h. Subsequently, the isolated protein was transferred to a nitrocellulose membrane at 40 V for 2 h. Each membrane was incubated at 4 °C with each primary antibody as follows: anti-ERK (Cell Signaling Technology Inc., Cambridge, MA, USA), anti-phospho-ERK (p-ERK) (Cell Signaling Technology Inc.), anti-p38 (Cell Signaling Technology Inc.), anti-p-p38 (Cell Signaling Technology Inc.), anti- IκB-α (Cell Signaling Technology Inc.), anti-p-IκB-α (Cell Signaling Technology Inc.), anti-C-kit (Abcam Com., Cambridge, UK), anti-NSE (Abcam Com.), PGP9.5 (Abcam Com.), anti-iNOS (Cell Signaling Technology Inc.), nNOS (Cell Signaling Technology Inc.), anti-Gα (Cell Signaling Technology Inc.), anti-mAChRs M2 (Alomone Labs, Jerusalem, Israel), anti-mAChRs M3 (Almone Labs), anti-PKC (Cell Signaling Technology Inc.), anti-p-PKC (Cell Signaling Technology Inc.), anti-PI3K (Cell Signaling Technology Inc.), and anti-p-PI3K(Cell Signaling Technology Inc.). Then, the membrane was washed with washing buffer (137 mM NaCl, 2.7 mM KCl, 10 mM Na_2_HPO_4_, and 0.05% Tween 20), and horseradish peroxidase (HRP)—conjugated goat anti-rabbit IgG (TransGen Biotech Co., Ltd., Beijing, China)—and incubated at room temperature for 1 h. The luminescence of the membrane was measured using the FluorChem^®^ FC2 Imaging system (Alpha Innotech Corporation, San Leandro, CA, USA). Finally, the density of each protein was quantified using the AlphaView Program (Cell Biosciences Inc., Santa Clara, CA, USA).

### 4.9. RT-qPCR Analysis

For collecting the total RNA of the mid colon, the tissue sample (40 mg) was homogenized using a Polytron PT-MR 3100D Homogenizer (Kinematica AG) with RNAzol (Tet-Test Inc., Friendswood, TX, USA). The pellets of the total RNAs were precipitated with isopropanol, dried in a SpeedVac Vacuum Concentrator, and dissolved with diethylpyrocarbonate (DEPC) water (20 µL). RNA concentration was measured with a NanoDrop^TM^ Spectrophotometer (Allsheng, Hangzhou, China), and complementary DNA (cDNA) was synthesized using an oligo-deoxythymidine (dT) primer (Thermo Fisher Scientific Inc.), 2′-deoxynuclease 5′-triphosphate (dNTP), and reverse transcriptase (SuperscriptII, Thermo Fisher Scientific Inc.). RT-qPCR was performed using a cDNA template (2 µL) and 2× Power SYBR Green (TOYOBO Co., Osaka, Japan).

The primer sequences of the target gene were as follows: MUC2, sense 5′-GCA CAT TCC TTC GCA TCT TAA A-3′ and anti-sense, 5′-AAA GCA AAG AAT GGA ACA GAA CAG AAA CTC-3′, MUC-1, sense 5′-CGCCA GCCTT GAGTT TGTTT-3′ and anti-sense, 5′-GAAGA AAGGA GCCCG AATGC-3′, Klf4, sense 5′-GGTGC AGCTT GCAGC AGTAA-3′ and anti-sense, 5′-AAGTC TAGGT CCAGG AGGTC GTT-3′, AQP8, sense 5′-TCGCT GGCAG TCACA GTGA-3′ and anti-sense, 5′-TCCAA ATAGC TGGGA GATCC A-3′, AQP3, sense 5′-GGTGG TCCTG GTCAT TGGAA-3′ and anti-sense, 5′-AGTCA CGGGC AGGGT TGA-3′, 5-HT 2AR, sense 5′-CCGGG AGCCT CTTGA TACAG-3′ and anti-sense, 5′-AGCCC CTCTC AAAGT CACAC A-3′, 5-HT 2BR, sense 5′-GCAGA TTTGC TGGTT GGATT G-3′ and anti-sense, 5′-GGCCA TATAG CCTCA AACAT GAT-3′, 5-HT 3AR, sense 5′-CTGAG GCCCT CCCAC ATCT-3′ and anti-sense, 5′-GGAAA GGAAC AAGGC CAACA-3′, 5-HT 3BR, sense 5′-TGCCG AGGAG TCTAG ATTGT ACCT-3′ and anti-sense, 5′-ACCCG ATGCT CCTGA TGGA-3′, ZO-1, sense 5′-CCTCC GTTGC CCTCA CAGTA-3′ and anti-sense, 5′-GGGCG CCCTT GGAAT G-3′, Claudin-1, sense 5′-CCCCG GAAAA CAACC TCTTA C-3′ and anti-sense, 5′-TGTCA CACAT AGTCT TTCCC ACTAG AA-3′, claudin-4, sense 5′-CGTGG CAAGC ATGCT GATTA-3′ and anti-sense, 5′-GTCGC GGATG ACGTT GTG-3′, Occludin, sense 5′-TTGAA GAGTG GGTTA AAAAT GTGTC T-3′ and anti-sense, 5′-TCAAC TCTTT CCGCA TAGTC AGAT-3′, p120, sense 5′-TGGAC CATGC GCTAC ACGCC-3′ and anti-sense, 5′-CCGAA GTTTC CGCCG GGCTT-3′, β-actin, sense 5′-ACGGC CAGGT CATCA CTATT G-3′ and anti-sense, 5′-CAAGA AGGAA GGCTG GAAAA GA-3′. RT-qPCR was performed for 40 cycles using the following sequence: denaturation at 95 °C for 15 s, annealing at 70 °C for 60 s, and extension at 70 °C for 60 s. Fluorescence intensities were measured at the end of the extension phase of each cycle. Threshold values for sample fluorescence intensities were set manually, and reaction cycles in which the PCR products exceeded the fluorescence intensity threshold during the exponential phase were considered threshold cycles (Ct). The expressions of each gene were quantified with respect to β-actin (the housekeeping gene) by comparing Ct values at a constant fluorescence intensity, as described by Livak and Schmittgen [64].

### 4.10. AChE Activity Assay

The activity of AChE was determined using an Acetylcholinesterase Assay Kit (DogenBio Co., Ltd., Seoul, Republic of Korea), according to the manufacturer’s instructions. Briefly, the collected mid colon tissues (20 mg) were homogenized using a Polytron PT-MR 310 D Homogenizer (Kinematica AG) in cold 0.1 M phosphate buffer (pH 7.5). Samples were centrifuged at 10,000× *g* for 5 min at 4 °C. The samples, standards, and Ach reaction mixture were incubated in a 96-well plate for 30 min at 37 °C, protected from the light. The absorbance of the mixture was read at 570 nm using the Molecular Devices VersaMax Plate Reader (Sunnyvale, CA, USA).

### 4.11. Measurement of NO Concentrations

The concentration of NO production was determined using the Griess reagent (Invitrogen Co., Ltd., Carlsbad, CA, USA) as in a previous study [65]. Briefly, the collected mid-colon tissues (40 mg) were homogenized using a Polytron PT-MR 310D Homogenizer (Kinematica AG) in 1× phosphate-buffered saline (PBS). After lysing these supernatants through a freeze and thaw cycle, these supernatants (100 µL) were mixed with 100 µL of modified Griess reagent (A:B = 1:1) in a 96-well plate. After incubation for 10 min, the absorbance of the mixture was read at 548 nm using the Molecular Devices VersaMax Plate Reader (SoftMax^®^ Pro Software, Sunnyvale, CA, USA).

### 4.12. Measurement of GI Hormones Concentrations

The concentrations of CCK and gastrin were quantified using an ELISA Kit (Cusabio Biotech Co., Ltd., Wuhan, China) according to the manufacturer’s instructions and a previous study [66]. Briefly, the collected mid colon tissues (40 mg) were homogenized using a Polytron PT-MR 3100 D Homogenizer (Kinematica AG) in 1× PBS. After lysing these supernatants through a freeze–thaw cycle, the specific antibody was mixed with the tissue lysate and incubated at 37 °C for 1 h. Subsequently, an HRP–streptavidin solution was added, and incubation was carried out for 1 h at 37 °C. This was followed by adding the 3,3′,5,5′-Tetramethylbenzidine (TMB) One-Step Substrate Reagent, followed by incubation of the mixture for 30 min at 37 °C. The reaction was terminated by adding the stop solution. Finally, the absorbance of the reaction mixture was read at 450 nm using the Molecular Devices VersaMax Plate Reader (Sunnyvale).

### 4.13. Statistical Analysis

Statistical analyses for all data were performed via a one-way analysis of variance (ANOVA) in SPSS release 27 for Windows (IBM SPSS, SPSS Inc., Chicago, IL, USA). And nonrandom associations between two categorical variables as post hoc analysis was evaluated using Fisher’s exact test and Tukey’s post hoc in SPSS. The results of the data are presented as means ± SDs, and statistical significance was accepted for *p* values < 0.05.

## 5. Conclusions

In this present study, we investigated whether Urd and AEtLP, which have been proven to be effective in Lop-induced constipation, could improve symptoms of C3-deficiency-induced constipation. To achieve this, the improvement in constipation-related parameters and the associated mechanisms were analyzed in 16-week-old C3 KO mice after Urd and AEtLP treatments. Significant improvements were seen in the stool parameters, histological structure, mucin secretion, water retention capacity, neuronal cell composition, ENS function, and GI hormone concentrations. Our results provide the first strong scientific evidence that C3-deficiency-induced constipation can be effectively treated by the administration of Urd and AEtLP. Also, our study provides comparative results on the therapeutic effects of Urd and AEtLP in two constipation models induced by two different triggers. Especially, the G protein is thought to play an important role in the therapeutic mechanism of Urd during Lop- and C3-deficiency-induced constipation due to the action of them being mediated by G proteins as a family of GPCRs. From these results, we believe that Urd and AEtLP may have potential in the treatment of constipation due to various causes. However, the differences in the animal species used to study Lop-induced constipation and C3-deficiency-induced constipation should be considered a limitation of our study. Therefore, the efficacy should be evaluated using the same species in an additional experiment to compare the results.

## Figures and Tables

**Figure 1 ijms-24-15757-f001:**
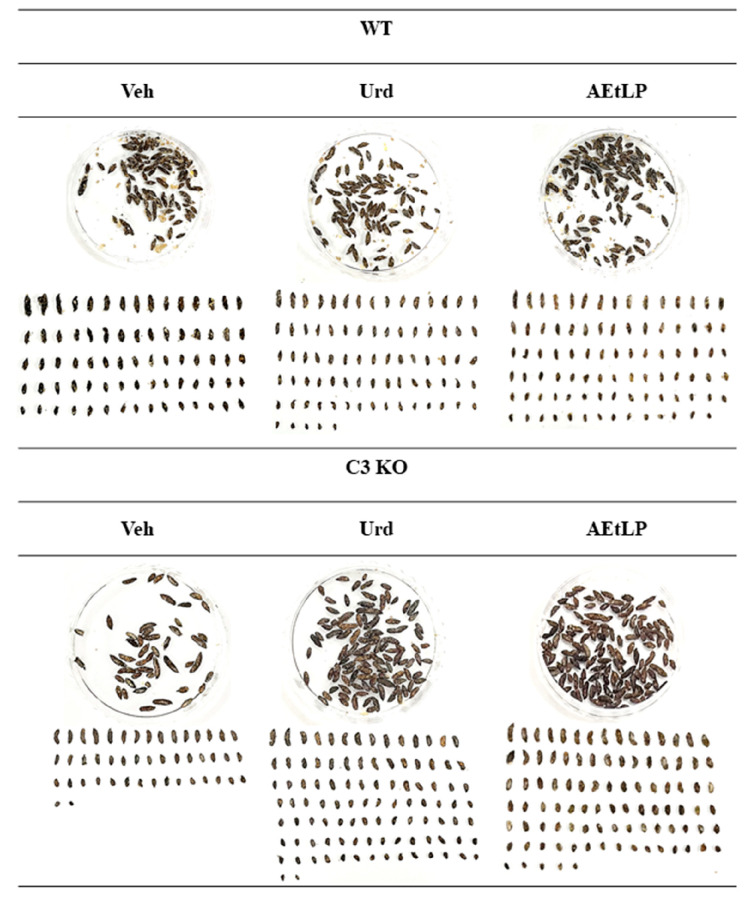
Stool number and morphology in C3 KO mice after the administration of Urd and AEtLP. The stools were arranged according to size and morphology and then photographed with a digital camera. The preparation of the total stools was performed on four to six mice per group and each parameter was analyzed twice for each sample. Abbreviations: WT, wild-type; C3 KO, complement component 3 knockout; Veh, vehicle; Urd, uridine; AEtLP, aqueous extract of *Liriope platyphylla* L.

**Figure 2 ijms-24-15757-f002:**
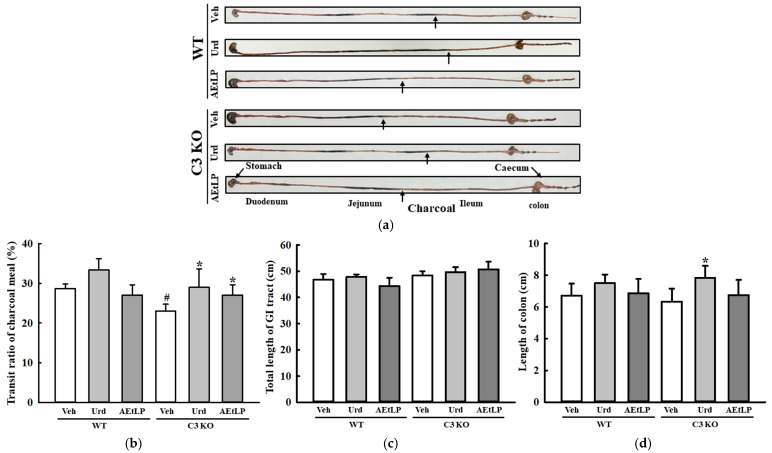
GI transit and length in C3 KO mice after administration of Urd and AEtLP. (**a**) Morphology of the GI tract. After collection from the stomach to the anus, the entire GI tract was arranged in a row, and the location of the charcoal meal was marked with arrows. (**b**) Transit ratio of charcoal meal. (**c**) Total length of GI tract. (**d**) Length of the colon. The preparation of the GI tract was performed on four to six mice per group, and each parameter was analyzed twice for each mouse. The data are presented as the mean ± SD. * *p* < 0.05 vs. Veh-treated group in the same mice. ^#^ *p* < 0.05 vs. WT mice in the same treated group. Abbreviations: WT, wild-type; C3 KO, complement component 3 knockout; GI, gastrointestinal tract; Veh, vehicle; Urd, uridine; AEtLP, aqueous extract of *Liriope platyphylla* L.

**Figure 3 ijms-24-15757-f003:**
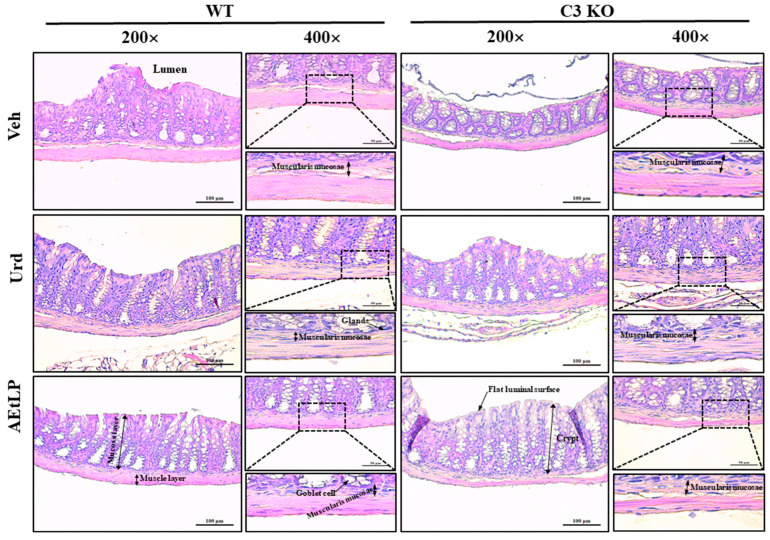
Histopathology of the mid colon in C3 KO mice after the administration of Urd and AEtLP. The dashed box was represented as an enlarged image at the bottom in the same column and each arrow indicated the name in the image. The H & E-stained tissue sections were prepared from tissue samples from four to six mice per group, and the pathological factors were analyzed twice for each stained tissue. Abbreviations: WT, wild-type; C3 KO, complement component 3 knockout; Veh, vehicle; Urd, uridine; AEtLP, aqueous extract of *Liriope platyphylla* L.; H & E, hematoxylin and eosin.

**Figure 4 ijms-24-15757-f004:**
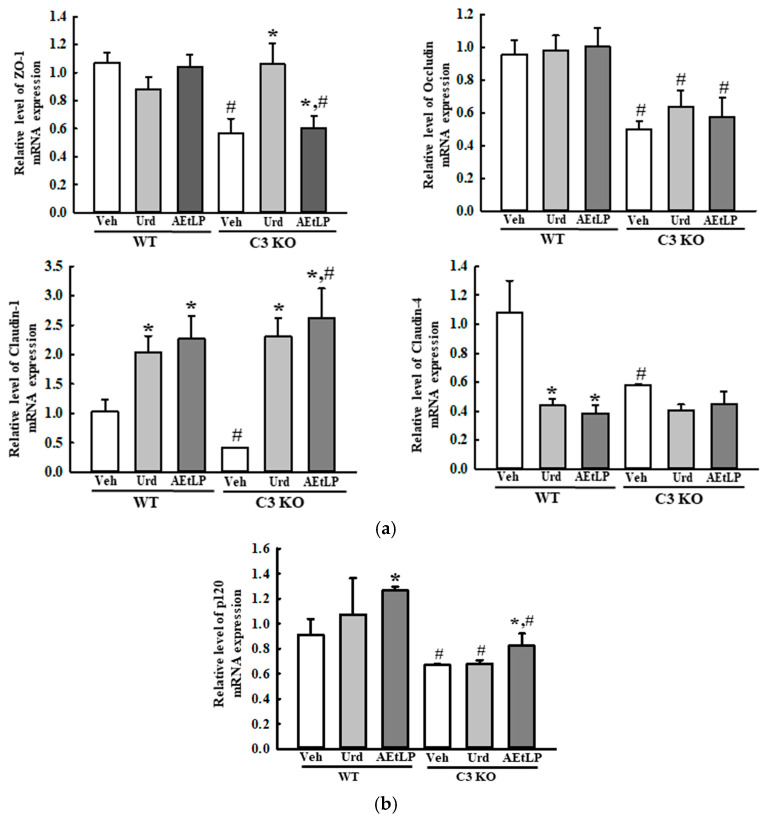
Expression of the junctional complexes in C3 KO mice after administration of Urd and AEtLP. (**a**) Transcription level of tight junctional complexes including ZO-1, occludin, claudin-1, and claudin-4. (**b**) Transcription level of the adherens junctional complexes including p120. The levels of these transcripts in the total mRNA of mid colons were measured via RT-qPCR using specific primers. The mRNA levels of five genes were calculated, based on the intensity of β-actin as an endogenous control. The preparation of total RNAs was performed on four to six mice per group. RT-qPCR analyses were assayed twice for each total RNA. The data are presented as the mean ± SD. * *p* < 0.05 vs. Veh-treated group in the same mice. ^#^ *p* < 0.05 vs. WT mice in the same treated group. Abbreviations: WT, wild-type; C3 KO, complement component 3 knockout; Veh, vehicle; Urd, uridine; AEtLP, aqueous extract of *Liriope platyphylla* L.; ZO-1, zonula occludens-1; RT-qPCR, quantitative real-time PCR.

**Figure 5 ijms-24-15757-f005:**
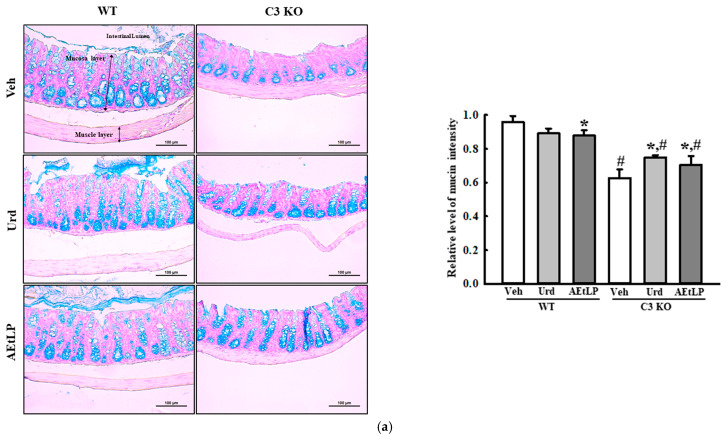
Mucin secretion in C3 KO mice after administration of Urd and AEtLP. (**a**) Detection of mucin in the Alcian-blue-stained colon section. After staining the colon section with Alcian blue solution, the blue color that represented the mucin level secreted from the crypt layer cells was observed at 200× magnification. The Alcian-blue-stained staining was performed on tissue sections taken from four to six mice per group. The color density was analyzed twice for each stained tissue. The scale bar represented 100 μm. (**b**) Transcription level of MUC-related genes. The mRNA levels of MUC1, MUC2, and Klf4 genes were calculated, based on the intensity of β-actin as an endogenous control. The preparation of total RNAs was performed on four to six mice per group, and the RT-qPCR analyses were performed twice for each total RNA. The data are presented as the mean ± SD. * *p* < 0.05 vs. Veh-treated group in the same mice. ^#^ *p* < 0.05 vs. WT mice in the same treated group. Abbreviations: WT, wild-type; C3 KO, complement component 3 knockout; Veh, vehicle; Urd, uridine; AEtLP, aqueous extract of *Liriope platyphylla* L.; MUC, mucin; Klf4, Kruppel-like factor 4; RT-qPCR, quantitative real-time PCR.

**Figure 6 ijms-24-15757-f006:**
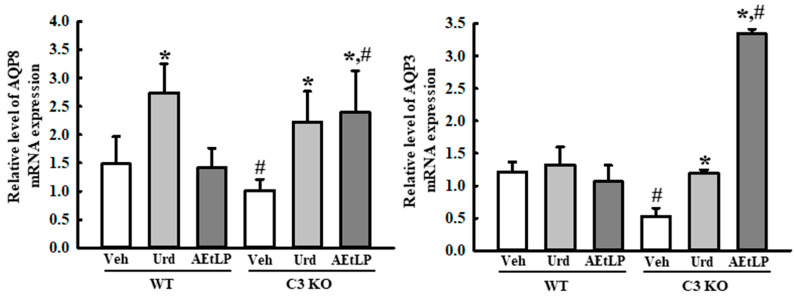
Expression of water channels in the C3 KO mice after administration of Urd and AEtLP. The levels of AQP3 and 8 transcripts in the total mRNA of the mid colons were measured via RT-qPCR using specific primers. The mRNA levels of two genes were calculated, based on the intensity of β-actin as an endogenous control. The analysis of total RNA levels was performed on four to six mice per group, and RT-qPCR analyses were performed twice for each total RNA. The data are presented as the mean ± SD. * *p* < 0.05 vs. Veh-treated group in the same mice. ^#^ *p* < 0.05 vs. WT mice in the same treated group. Abbreviations: WT, wild-type; C3 KO, complement component 3 knockout; Veh, vehicle; Urd, uridine; AEtLP, aqueous extract of *Liriope platyphylla* L.; AQP, aquaporin; RT-qPCR, quantitative real-time PCR.

**Figure 7 ijms-24-15757-f007:**
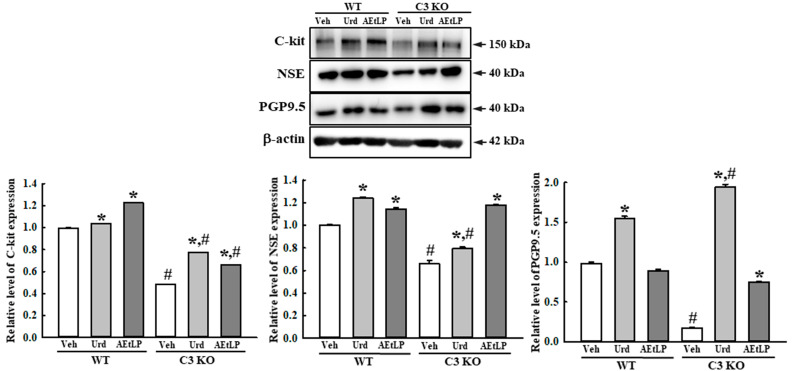
Expression levels of C-kit, NSE, and PGP9.5 in the C3 KO mice after administration of Urd and AEtLP. After collecting the mid colons from the C3 KO mice, the expression levels of C-kit, NSE, and PGP9.5 were assessed in tissue homogenates using the specific primary antibody and densitometry. The tissue homogenates were prepared from four to six mice per group, and the Western blot was analyzed twice for each sample. The level of each protein was normalized to β-actin. The data are presented as the mean ± SD. * *p* < 0.05 vs. Veh-treated group in the same mice. ^#^ *p* < 0.05 vs. WT mice in the same treated group. Abbreviations: WT, wild-type; C3 KO, complement component 3 knockout; Veh, vehicle; Urd, uridine; AEtLP, aqueous extract of *Liriope platyphylla* L.; C-kit, receptor tyrosine kinase kit; NSE, neuron-specific enolase; PGP9.5, protein gene product 9.5.

**Figure 8 ijms-24-15757-f008:**
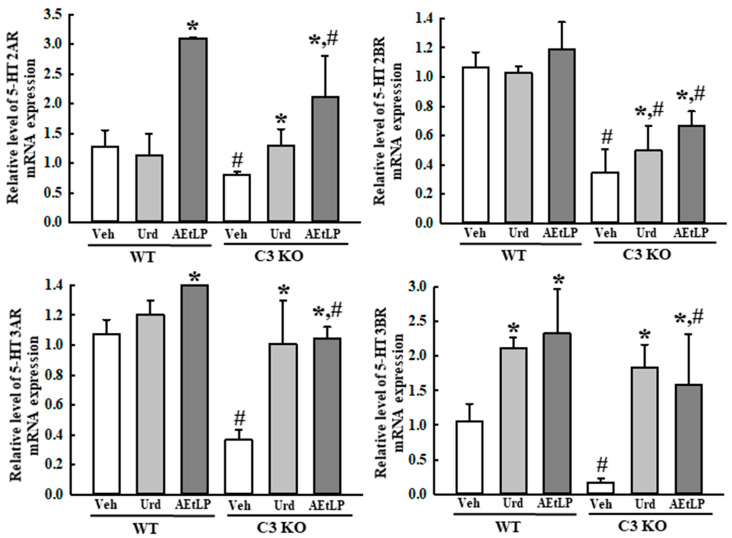
Expression of 5-HT receptors in the C3 KO mice after administration of Urd and AEtLP. The levels of 5-HT 2AR, 2BR, 3AR, and 3BR transcripts in the total mRNA of mid colons were measured via RT-qPCR using specific primers. The mRNA levels of four genes were calculated, based on the intensity of β-actin as an endogenous control. The total RNA analysis was performed on four to six mice per group, and the RT-qPCR analyses were assayed twice for each total RNA. The data are presented as the mean ± SD. * *p* < 0.05 vs. Veh-treated group in the same mice. ^#^ *p* < 0.05 vs. WT mice in the same treated group. Abbreviations: WT, wild-type; C3 KO, complement component 3 knockout; Veh, vehicle; Urd, uridine; AEtLP, aqueous extract of *Liriope platyphylla* L.; 5-HT, 5-hydroxytryptamine; RT-qPCR, quantitative real-time PCR.

**Figure 9 ijms-24-15757-f009:**
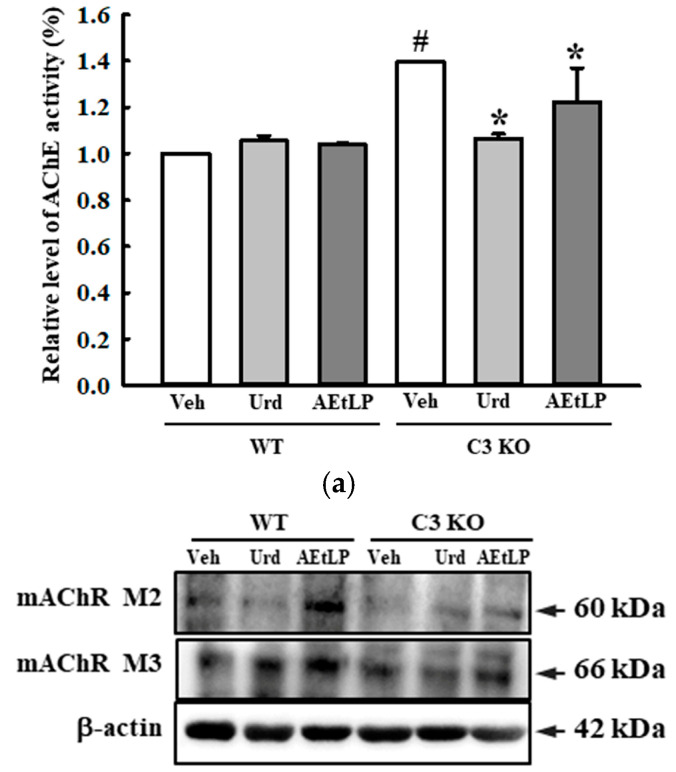
Expression levels of key members of the mAChRs’ downstream signaling pathway in C3 KO mice after administration of Urd and AEtLP. (**a**) Activity of AChE. After homogenizing the mid-colon tissue, AChE activity was measured using an Acetylcholinesterase Assay Kit that could detect as little as 0.01 mU AChE in a 100 µL assay sample (0.1 mU/mL). The tissue homogenates were prepared from four to six mice per group, and AChE activity was assayed twice for each sample. (**b**) Expression levels of mAChR M2 and M3. (**c**) Expression level of Gα. (**d**) Phosphorylation levels of PKC and PI3K. (**e**) Phosphorylation level of MAPK members. After collecting the mid colons from C3 KO mice, the expression levels of these proteins were assessed in the tissue homogenates using the specific primary antibody and densitometry. The tissue homogenates were prepared from four to six mice per group, and the Western blot was analyzed twice for each sample on different mice samples. The level of each protein was normalized to β-actin. The data are presented as the mean ± SD. * *p* < 0.05 vs. Veh-treated group in the same mice. ^#^ *p* < 0.05 vs. WT mice in the same treated group. Abbreviations: WT, wild-type; C3 KO, complement component 3 knockout; Veh, vehicle; Urd, uridine; AEtLP, aqueous extract of *Liriope platyphylla* L.; AChE, acetylcholinesterase; mAChR, muscarinic acetylcholine receptors; Gα, G-protein alpha subunit; PKC, protein kinase C; PI3K, phosphoinositide 3-kinase; ERK, extracellular signal-regulated kinase; IκB-α, nuclear factor kappa B inhibitor alpha.

**Figure 10 ijms-24-15757-f010:**
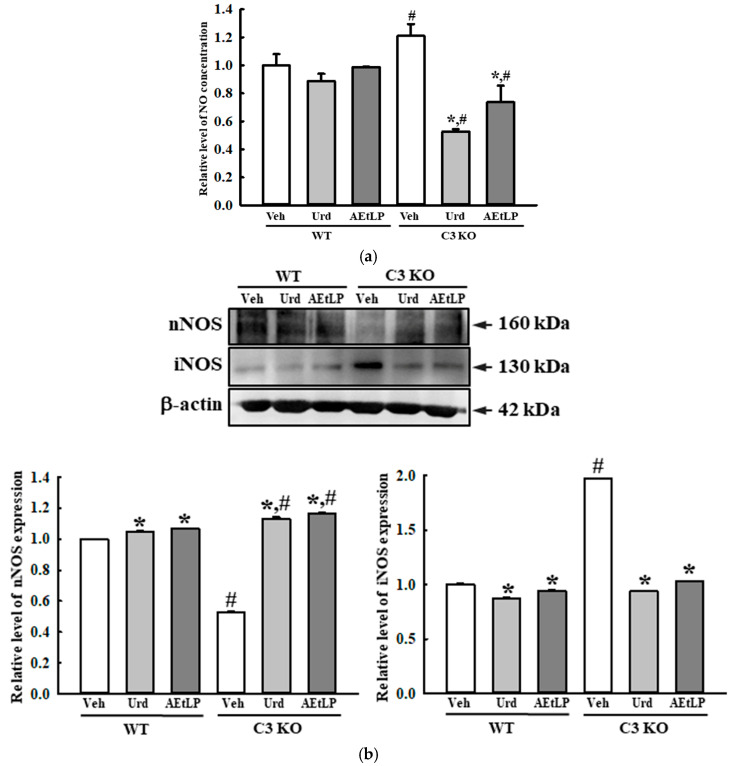
Concentration of NO and expression of nNOS and iNOS in the C3 KO mice after administration of Urd and AEtLP. (**a**) Concentration of NO in the mid colon. NO concentration in the tissue homogenates was determined using the Griess reagent. The tissue homogenates were prepared from four to six mice per group, and the NO assay was conducted twice for each sample. (**b**) Expression levels of nNOS and iNOS in the mid colon. After collecting the mid colons from C3 KO mice, the expression levels of these proteins were assessed in tissue homogenates using the specific primary antibody and densitometry. The tissue homogenates were prepared from four to six mice per group, and the Western blot was analyzed twice for each sample on different mice samples. The level of each protein was normalized to β-actin. The data are presented as the mean ± SD. * *p* < 0.05 vs. Veh-treated group in the same mice. ^#^ *p* < 0.05 vs. WT mice in the same treated group. Abbreviations: WT, wild-type; C3 KO, complement component 3 knockout; Veh, vehicle; Urd, uridine; AEtLP, aqueous extract of *Liriope platyphylla* L.; NO, nitric oxide; iNOS, inducible nitric oxide synthase; nNOS, neuronal NOS.

**Figure 11 ijms-24-15757-f011:**
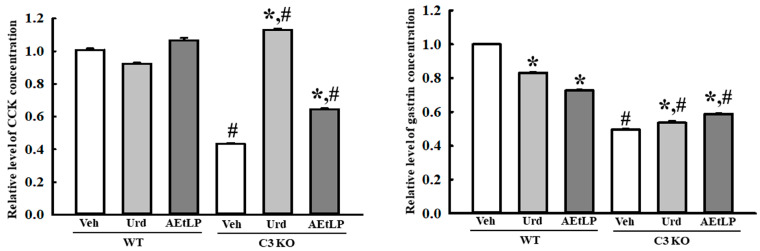
Concentrations of GI hormones in the C3 KO mice after administration of Urd and AEtLP. The concentrations of CCK and gastrin were measured in the mid colon homogenate using an ELISA. The minimum detectable concentration of each kit was 0.1–1000 pg/mL of CCK and 0.312–20 pg/mL of gastrin. The tissue homogenates were prepared from four to six mice per group, and the hormone levels were analyzed twice for each sample on different mice samples. The data are presented as the mean ± SD. * *p* < 0.05 vs. Veh-treated group in the same mice. ^#^ *p* < 0.05 vs. WT mice in the same treated group. Abbreviations: WT, wild-type; C3 KO, complement component 3 knockout; Veh, vehicle; Urd, uridine; AEtLP, aqueous extract of *Liriope platyphylla* L.; ELISA, enzyme-linked immunosorbent assay; CCK, cholecystokinin; GI, gastrointestinal.

**Table 1 ijms-24-15757-t001:** Comparison of excretion and feeding parameters between WT and C3 KO mice after administration of uridine and AEtLP.

Categories	WT			C3 KO		
	Veh	Urd	AEtLP	Veh	Urd	AEtLP
Stool number (ea)	105.3 ± 7.6	95.3 ± 1.2	131 ± 4.6 *	53 ± 10 ^#^	121 ± 19.8 *^,#^	91.7 ± 4.9 *^,#^
Stool weight (g)	0.9 ± 0.3	0.9 ± 0.3	0.9 ± 0.2	0.5 ± 0.1 ^#^	0.9 ± 1.1	1.3 ± 0.2 *^,#^
Stool water contents (%)	20 ± 2.2	22.9 ± 3	43.4 ± 7.3 *	8 ± 1.2 ^#^	20.8 ± 1.6 *	35.8 ± 4.8 *^,#^
Urine volume (mL)	1 ± 0.09	1.3 ± 0.05	1.35 ± 0.05	0.95 ± 0.07	1.25 ± 0.07	0.8 ± 0.05
Food intake (g)	8 ± 1	12.4 ± 2.3	9.5 ± 0.5	14.1 ± 0.1	10.5 ± 0.7	16 ± 0.05
Water consumption (mL)	6 ± 1.4	5.8 ± 0.6	6.8 ± 1	6.7 ± 0.6	6.4 ± 0.6	7.7 ± 1.5

The data are presented as the mean ± SD. * *p* < 0.05 vs. Veh-treated group in the same mice. ^#^ *p* < 0.05 vs. WT mice in the same treated group. Abbreviations: WT, wild-type; C3 KO, complement component 3 knockout; Veh, vehicle; Urd, uridine; AEtLP, aqueous extract of *Liriope platyphylla* L.

**Table 2 ijms-24-15757-t002:** Comparison of histological parameters between WT and C3 KO mice after administration of Urd and AEtLP.

Categories	Feature (Unit)	WT			C3 KO		
		Veh	Urd	AEtLP	Veh	Urd	AEtLP
Crypt	Length (µm)	140.2 ± 4.5	145.7 ± 8.9	147.8 ± 2.9	78.3 ± 2.9 *	136.2 ± 11.2 ^#^	170.6 ± 6.7 *^,#^
Flat luminal surface	Thickness (µm)	14.1 ± 1.7	14.3 ± 1.5	17.02 ± 1.4 *	8.7 ± 1.0 *	15 ± 1.5 ^#^	15 ± 0.7 ^#^
Muscle layer	Thickness (µm)	58.2 ± 1.7	56.2 ± 2.3	45.8 ± 2.4 *	34.9 ± 0.9 *	21.2 ± 0.9 *^,#^	34.9 ± 1.8 *
Mucosa layer	Thickness (µm)	192.8 ± 0.72	187.7 ± 9.8	172.2 ± 3.8 *	107.7 ± 9.2 *	115.2 ± 4.1 *	149.5 ± 2.6 *^,#^
Epithelial cells	Enterocyte, goblet cells, and paneath cells	Regular arrangement	Regular arrangement	Regular arrangement	Irregular arrangement Cell-shaped destruction Decrease in cell number	Cell-shaped recovery Recovery of cell number	Cell-shaped recovery Recovery of cell number
Muscularis mucosae	Thickness (µm)	8.18 ± 0.8	7.8 ± 0.5	7.8 ± 1.1	14.7 ± 0.8 *	9.8 ± 1.0 ^#^	15.31 ± 3.1 *

The H & E-stained tissue sections were prepared from tissue samples from four to six mice per group, and the pathological factors were analyzed twice for each stained tissue. The data are presented as the mean ± SD. * *p* < 0.05 vs. Veh-treated group in the same mice. ^#^ *p* < 0.05 vs. WT mice in the same treated group. Abbreviations: WT, wild-type; C3 KO, complement component 3 knockout; Veh, vehicle; Urd, uridine; AEtLP, aqueous extract of *Liriope platyphylla* L.

**Table 3 ijms-24-15757-t003:** Comparison of Urd and AEtLP laxative effect between LOP-induced constipation and C3-deficiency-induced constipation.

Categories	Alteration Rate after Treatment			
	Urd		AEtLP	
	Lop- Induced Constipation	C3-Deficiency-Induced Constipation	Lop- Induced Constipation	C3-Deficiency-Induced Constipation
Stool number (ea)	85.2 ± 33%	129.2 ± 7%	67.8 ± 16%	74.9 ± 26%
Stool weight (g)	105.7 ± 49%	82.2 ± 17%	96.4 ± 0.4%	161.7 ± 13%
Stool water contents (%)	110.5 ± 12%	159.5 ± 20%	122 ± 73%	339 ± 16%
Gastrointestinal motility (%)	ND	36 ± 6%	ND	27 ± 3%
Intestine length (µm)	ND	2.7%	ND	4.8 ± 2%
Mucosa layer (µm)	275.3 ± 38%	7.8 ± 5%	10.3 ± 8.7%	39.8 ± 10%
Muscle layer (µm)	221.3 ± 25%	−39.8 ± 1%	124 ± 24%	ND
Flat luminal surface thickness (μm)	39.3 ± 4%	72.6 ± 3%	200 ± 0%	73.3 ± 12%
AQP8 level (%)	513.8 ± 28%	118.8 ± 18%	ND	126.8 ± 47%
MUC2 level (%)	150 ± 0%	140.2 ± 4%	ND	220.8 ± 15%
mAChRM2 level (%)	254.8 ± 43%	28.6 ± 7%	−20 ± 4%	50 ± 7%
mAChRM3 level (%)	237.8 ± 40%	−3 ± 4.8%	−7.7 ± 0.3%	60.7 ± 8%
PI3K level (%)	−43.6 ± 3%	−11.7 ± 2%	−47.7 ± 4%	−29.4 ± 1%
PKC level (%)	−92.8 ± 0.5%	−23 ± 1%	−37.9 ± 0.4%	−64 ± 0.8%

The data are presented as the mean ± SD. Abbreviations: WT, wild-type; C3 KO, complement component 3 knockout; Veh, vehicle; Urd, uridine; AEtLP, aqueous extract of *Liriope platyphylla* L.; LOP, loperamide; AQP, aquaporin; MUC, mucin; mAChR, muscarinic acetylcholine receptors; PI3K, phosphoinositide 3-kinase; PKC, protein kinase C; ND, Not detected.

## Data Availability

Data openly available in a public repository that issues datasets with DOIs.

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
