# Peer review of "Complement C3-Deficiency-Induced Constipation in FVB/N-C3em1Hlee/Korl Knockout Mice Was Significantly Relieved by Uridine and Liriope platyphylla L. Extracts"

_ijms, 2023, doi:10.3390/ijms242115757_

Round 1
Reviewer 1 Report
Comments and Suggestions for Authors
In general, the Introduction seems to be sufficient, but I would recommend expanding the part on the relevance of the study. Probably some statistics on the prevalence of constipation or related effects on other organ systems would be appropriate. The need to look for new drugs is also not very clear. Are there currently not enough medicines? How many officially registered (for example, in their own country) drugs can the authors name? Are they ineffective? This question remained unanswered for me. The choice of two substances also does not seem reasonable. Is one of them a positive control? If yes, it should be described. If not, was a positive control used?
Discussion of the obtained data is completely insufficient. In the IJMS, I would like to see a reasoning about the proposed mechanism of action, taking into account the data received and the information available.
However, the authors only compare the effects with their own data in another model. Obviously this needs to be fixed.
Moreover, the authors do not discuss the composition of the extract in connection with its action at all. A large number of references are given to studies of the activity of the extract of Liriope platyphylla (and therefore the novelty is not so great), but there is no information about what substances in its composition could act this way.
The extract of Liriope platyphylla caused a significant increase in the stool number and water content in wild-type animals, i.e. control. The authors need to discuss and explain this effect, which without it looks non-specific.
Moreover, the Conclusions do not contain any specific conclusions about the action of the studied substances.
I also see that the authors have cited a lot of their work compared to others. This needs to be corrected or justified.
Therefore, I do not see that the study was planned and the results were analyzed. In this regard, the manuscript in this form cannot be recommended for publication and needs to be carefully rewritten.
Reviewer 2 Report
Comments and Suggestions for Authors
I have some critical comments:
INTRODUCTION
• The Introduction starts with a general description of constipation, its causes, and the relevance of studying it. However, transitioning to the specific focus on complement C3 deficiency-induced constipation is a bit abrupt. It might be helpful to use clearer subheadings or transition sentences to guide the reader through the shift in focus from general constipation to the specific research problem.
• The Introduction does a good job of presenting the background and context of the study, including the recent investigations into complement C3 deficiency as a cause of chronic constipation. However, it could be strengthened by explicitly stating the research gap or question that this study aims to address. What is the specific hypothesis or objective of this research, and how does it fit into the existing body of knowledge on this topic? Providing a straightforward research question or hypothesis would enhance the overall focus of the Introduction. Please clearly state the clinical impressions of your study in the last paragraph of the Introduction. What problems remain unanswered? What are questions responding to?
METHODS
• While the methods are generally well-described, some steps could benefit from more explicit and clear instructions. For example, in section 4.1, where Urd and AEtLP preparation is mentioned, it would be helpful to provide more details on the specific procedures used for extraction, such as the duration, temperature, and any specific solvents used.
• When referring to methods or procedures used in previous studies (e.g., in sections 4.2, 4.3, and 4.12), it is crucial to provide proper citations for these methods. The reader should be able to access the referenced studies to better understand these procedures.
• While the study mentions statistical analysis in section 4.13, providing more information about the specific statistical tests used and how the data were analyzed would be beneficial. This includes mentioning any software packages or tools used for statistical analysis, such as the version of SPSS, and specifying the exact statistical tests performed.
DISCUSSION
• What were your strengths and limitations?
Minor editing of English language is required.
Reviewer 3 Report
Comments and Suggestions for Authors
Comments in attached file.

Quality of English is satisfactory.
Reviewer 4 Report
Comments and Suggestions for Authors
This manuscript titled “T Complement C3 deficiency-induced constipation in FVB/N-C3em1Hlee/Korl knockout mice was significantly relieved by uridine and Liriope platyphylla L. extracts”. The comments for this manuscript are as follows:
The information in the entire manuscript seems to be quite sufficient and complete, and there are no mistakes in the reference format. The format of superscripts and subscripts is also correct, which shows the carefulness of the authors. But the reviewer only asked two questions for the authors to respond to me.
1. In the manuscript, the authors only mentions the extraction process of Liriope platyphylla L., but it is not complete. How long does it take to extract 600 g of Liriope platyphylla L. at 60°C? What is its yield? None of the authors stated. Most importantly, a series of experiments were conducted on warm water extracts without any chemical analysis of the extracts, which may not be of much help to future researchers. In fact, plant extracts are quite complex, and the chemical composition of the materials obtained every year may vary. Is it meaningful to conduct molecular and cell experiments before the chemical composition is clear? The same question also is for the MAPKs measured by the authors. If the authors cannot provide further information about this extract, I think this manuscript will be of little significance to the readers.
2. The PKC level in Table 3 decreased by 92.8%. Why is there no explanation in the article?
I decided it should be a major revision.
Round 2
Reviewer 1 Report
Comments and Suggestions for Authors
The authors did not highlight any changes in the manuscript, so it is extremely difficult to track them. This is against the rules for authors and is disrespectful to the reviewer.
The manuscript may be published if the editor decides to do so.
Reviewer 2 Report
Comments and Suggestions for Authors
Although the authors did not highlight the revisions, I could find them! Anyway, thanks for the changes.
Reviewer 3 Report
Comments and Suggestions for Authors
The paper is now much improved and is suitable for publication.
Comments on the Quality of English LanguageQuality of English is satisfactory.